# Clinical Significance of Tumour-Infiltrating B Lymphocytes (TIL-Bs) in Breast Cancer: A Systematic Literature Review

**DOI:** 10.3390/cancers15041164

**Published:** 2023-02-11

**Authors:** Brian M. Lam, Clare Verrill

**Affiliations:** 1Department of Oncology, University of Oxford, Oxford OX3 9DU, UK; 2Nuffield Department of Surgical Sciences, University of Oxford, John Radcliffe Hospital, Oxford OX3 9DU, UK; 3Oxford NIHR Biomedical Research Centre, University of Oxford, Oxford OX3 9DU, UK; 4Department of Cellular Pathology, Oxford University Hospitals NHS Foundation Trust, John Radcliffe Hospital, Oxford OX3 9DU, UK

**Keywords:** tumour-infiltrating B lymphocytes, breast cancer, tumour microenvironment

## Abstract

**Simple Summary:**

Breast cancer is the most frequently diagnosed cancer among women worldwide. Although there have been significant advances in the past decade in our understanding of breast cancer biology and the choice of treatment, mortality remains high, especially among certain subgroups of patients. The advent of onco-immunology as a burgeoning speciality of precision medicine designed to heighten the antitumor response of the immune system in cancers like melanoma has not been associated with impressive results in breast cancer. Most research efforts have focused only on T lymphocytes with little regard to other potentially important players, including B lymphocytes. This review aims to summarise studies that have assessed the clinical significance of tumour-infiltrating B lymphocytes in breast cancer and explore future research directions that may shed further light on understanding the role of B lymphocytes in breast cancer.

**Abstract:**

Although T lymphocytes have been considered the major players in the tumour microenvironment to induce tumour regression and contribute to anti-tumour immunity, much less is known about the role of tumour-infiltrating B lymphocytes (TIL-Bs) in solid malignancies, particularly in breast cancer, which has been regarded as heterogeneous and much less immunogenic compared to other common tumours like melanoma, colorectal cancer and non-small cell lung cancer. Such paucity of research could translate to limited opportunities for this most common type of cancer in the UK to join the immunotherapy efforts in this era of precision medicine. Here, we provide a systematic literature review assessing the clinical significance of TIL-Bs in breast cancer. Articles published between January 2000 and April 2022 were retrieved via an electronic search of two databases (PubMed and Embase) and screened against pre-specified eligibility criteria. The majority of studies reported favourable prognostic and predictive roles of TIL-Bs, indicating that they could have a profound impact on the clinical outcome of breast cancer. Further studies are, however, needed to better define the functional role of B cell subpopulations and to discover ways to harness this intrinsic mechanism in the fight against breast cancer.

## 1. Introduction

Breast cancer is the most common type of cancer in the UK [1]. With over 2 million new cases being diagnosed across the world in 2020, it has now surpassed lung cancer as the most commonly diagnosed cancer among women [2]. Despite impressive advances in therapeutic modalities, breast cancer remains the leading cause of cancer-related deaths among women worldwide, accounting for over 15% of female cancer mortality [2]. Certain subtypes, including HER2-enriched and triple-negative breast cancer (TNBC), are notoriously aggressive, carrying significant risks of early spread and tumour recurrence [3]. Responses to conventional chemotherapy and radiotherapy are usually not long-lasting, and efficacy is very limited in metastatic and relapsed diseases [4]. Much effort has therefore been made to continuously advance our understanding of the pathophysiology of breast cancer, with the hope of identifying novel strategies to improve survival outcomes of this malignancy.

In the last two decades, cancer treatment has been revolutionised by our increased understanding of the tumour microenvironment (TME). It has now become clear that the initiation and progression of cancer is a complex and dynamic process that involves not only the intrinsic genetic aberrations of the tumour tissue but also its interactions with the local microenvironment [5] constituted by the extracellular matrix, cancer-associated fibroblasts, infiltrating inflammatory cells, vascular and lymphatic networks, as well as a variety of cytokines and biomolecules [6]. The concept of cancer immunoediting [7], introduced by Dunn and Schreiber in 2002, further describes the process of malignant neoplastic progression on the basis of tumour and immune cell interactions [8]. Such knowledge of our immune system’s capacity to recognise, target and destroy cancer cells has sparked considerable interest in harnessing this intrinsic mechanism to fight against cancer.

Standing at the very heart of this tumour-immune system interaction are the tumour-infiltrating lymphocytes (TILs), which are a mixture of T cells (TIL-Ts), B cells (TIL-Bs) and natural killer (NK) cells observed to be nested in and around the neoplastic cells [9]. Together with the relatively minor populations of macrophages and dendritic cells (DCs) [10], TILs exert influence on the behaviour of cancer cells and play a deterministic role in the course of cancer development through a myriad of adaptive immune responses, which are believed to be initiated by tumour-specific peptides known as neoantigens or tumour-associated antigens (TAAs) that have arisen as a result of expressed somatic cancer mutations [11], and are presented to effector cells of the immune system by complexing with major histocompatibility complex (MHC) proteins. Most of the immuno-oncology research to date has focused on cell-mediated immunity (CMI). Under the classic model, the immune system plays a dichotomous role in modulating tumour progression, where cytotoxic T cells, helper T cells, NK cells and DCs contribute to the anti-tumour response, and other cells like Foxp3+ regulatory T cells (Tregs) and myeloid-derived suppressor cells (MDSCs) have a pro-tumorigenic effect that promotes cancer growth and invasion by producing cytokines sustaining angiogenesis and stimulating tumour cell proliferation [12].

Despite being one of the main constituents of the TME, the roles of TIL-Bs in cancers are not well characterised, and studies examining their prognostic significance are much more limited [13,14]. In retrospective analyses of mastectomy specimens, a progressive increase in the density of B-cell infiltrate was observed as mammary glands transition from normal tissue through usual and atypical ductal hyperplasia to ductal carcinoma in-situ (DCIS) and infiltrating ductal carcinoma [15,16,17]. It is believed that antigens expressed on breast cancer cells are the most important factor driving B cell proliferation and infiltration into the tumour tissue [18]. In addition, cancer cells that produce B cell chemoattractants like C-X-C motif chemokine ligand 13 (CXCL13) may also play an important role in attracting B cells to the microenvironment [19].

B cells are known to play a critical role in the development of humoral immune responses. They arise from the bone marrow and, upon stimulation by antigens, progress through a series of developmental stages to eventually differentiate into memory B cells and plasma cells (PCs). Throughout these stages of development, B cells express different markers on their cell surface, which allow them to be identified immunohistochemically (Table 1). Recent studies in human cancers revealed that TIL-B phenotypes include naïve B cells, naive activated B cells, germinal centre (GC) B cells, memory B cells, as well as PCs and their intermediates [20]. Recent single-cell RNA sequencing (scRNA-seq) studies of breast cancer tissue even reported up to 13 TIL-B phenotypes, spanning the entire continuum from naïve B cells to PCs [21]. It is believed that mature B cells within the TME could be activated by neoantigens or TAAs on cancer cells to differentiate into PCs, which could produce immunoglobulins (Ig) [22] and mediate tumour cell elimination through cell lysis by NK cells (antibody-dependent cell cytotoxicity, ADCC), complement activation (complement-mediated cytotoxicity) and macrophage phagocytosis. Analysis of Ig heavy chain genes in TIL-Bs revealed evidence of clonal expansion, somatic hypermutation, affinity maturation and class switching, reminiscent of GC reactions in an active humoral immune response [23]. In addition, activated TIL-Bs can theoretically act as professional antigen-presenting cells (APCs) and present epitopes to TIL-Ts to induce tumour antigen-specific T-cell responses [24]. In a recent study on non-small cell lung cancer (NSCLC), TIL-Bs directly isolated from cancer samples were shown to stimulate autologous CD4+ TIL-Ts in vitro, providing empirical evidence to support this mechanism [25].

Complicating the picture, however, are the reported immune-inhibitory properties of TIL-Bs. More specifically, there is a subset of B cells that is known to suppress T cell-mediated anti-tumour response by secreting suppressive cytokines like interleukin-10 (IL-10) and transforming growth factor-beta (TGF-β), which can suppress macrophage and NK cell activities, promote the differentiation of naïve CD4+ T cells into regulatory T cells (Tregs) and upregulate the expression of immune checkpoint molecules like programmed death protein-1 (PD-1) and programmed death-ligand 1 (PD-L1) [26]. Signal transducer and activator of transcription 3 (STAT3) signalling also mediates vascular endothelial growth factor (VEGF) to promote angiogenesis and metastasis [27,28].

Given the phenotypic and functional diversities of B cells, their role in tumour progression is conceivably complex, and their clinical significance in breast cancer as a disease with high heterogeneity is, not surprisingly, controversial. To address this area of confusion, a systematic review was conducted, which aims to summarise and critically evaluate findings from primary studies that have specifically assessed the prognostic and/or predictive significance of TIL-Bs in breast cancer and identify gaps in knowledge that could help inform future research needs. Clarifying the clinical significance of TIL-Bs could enhance our understanding of B cells in the anti-tumour immune response and potentially open new avenues for the development of novel immunotherapies for breast cancer.

## 2. Materials and Methods

### 2.1. PRISMA Statement

The protocol of this review has consulted the PRISMA (Preferred Reporting Items for Systematic Reviews and Meta-Analyses) statement, a guideline developed in 2009 and updated in 2020 to help homogenise how systematic reviews are conducted and improve the completeness of their reporting [29]. This review has not been registered.

### 2.2. Search Strategy

A systematic literature search was conducted within two electronic databases: PubMed/MEDLINE and EMBASE, for original articles which met our inclusion criteria and were published from January 2000 to April 2022. Keywords, their synonyms and Boolean operators were used in the search process, as shown in Figure 1 below. Citations in selected papers and related articles suggested by PubMed were also reviewed to identify more potentially relevant studies.

### 2.3. Selection Process

The search results were then reviewed for duplicates, and the remaining articles were screened against specified inclusion and exclusion criteria, as detailed below. Titles and abstracts that were deemed relevant on preliminary screening were examined further at the full-text level. The same applies to records whose suitability was equivocal at either the title or abstract level in order to avoid the omission of potentially relevant or informative studies.

Studies for inclusion were those that:Specifically scored TIL-Bs in the tumour bed.Evaluated the association between TIL-Bs and clinical outcomes in breast cancer, such as disease-specific survival (DFS), progression-free survival (PFS), overall survival (OS) and response to chemotherapy.

Studies for exclusion were those that:Were not pertinent to breast cancer.Evaluated tils as a single population and did not specifically assess TIL-Bs.Had a sample size below 50.Were non-empirical or secondary resources, including reviews, commentaries and academic textbooks.Were not written in the English language.Were not pertinent to humans.

### 2.4. Data Extraction

Data from eligible publications were extracted in a systematic manner. The following data items were retrieved from each article: the first author’s name, year of publication, country of the cohorts studied, sample size, breast cancer subtype, clinicopathological features of the tumour, type of treatment received, duration of follow-up and clinical outcomes or prognostic parameters reported. In addition, details related to the methods of TIL-Bs detection were extracted, including the immunohistochemical markers and antibody clones used, localisation or compartmentalisation of B cells within the tumour bed (intratumoural vs. stromal), scoring methodology and cutoff value.

## 3. Results

### 3.1. Characteristics of Studies

#### 3.1.1. Overview

The study selection process is illustrated in Figure 2. A total of 1191 articles were identified after the initial search. After the exclusion of the duplicate records (n = 354), a total of 837 unique articles were screened at the title and abstract levels, warranting 65 studies for full-text review. At this level, 51 articles were removed for not satisfying the eligibility criteria (e.g., no data concerning TIL-Bs or clinical outcomes). Finally, a total of 14 studies [30,31,32,33,34,35,36,37,38,39,40,41,42,43] were included in the current systematic review. The main characteristics of each eligible study can be found in Appendix A.

The sample size of each study ranged from 80 to 1470, adding up to a total of 4105 patients. Most studies (n = 12) enrolled >100 patients, and one study even enrolled >1000 patients. All studies included were retrospective analyses conducted between 2011 and 2021 in cohorts from Europe (n = 7), Asia (n = 4), North America (n = 2) and Africa (n = 1), evaluating both invasive (n = 13) and in-situ breast cancer (n = 1). The median length of follow-up ranged from 35 months to 266 months. Histological and molecular subtypes of invasive breast cancer were specified in some studies, including TNBC (n = 7), HER2+ (n = 2), invasive ductal carcinoma (IDC) (n = 5) and invasive lobular carcinoma (ILC) (n = 1). The majority of studies (n = 13) used immunohistochemistry (IHC) to detect TIL-Bs, whereas one study used immunofluorescence (IF), and the remaining one relied on the morphological assessment of haematoxylin and eosin (H&E)-stained sections alone for evaluating TIL-Bs abundance. Among studies that utilised IHC to detect TIL-Bs, eight of them used CD20 alone to stain B cells as a general population, whilst four of them used additional stains, including CD38 (n = 2) and/or CD138 (n = 3) for further subtyping. Eight studies specified the area(s) of assessment, with three of them focusing only on stromal lymphocytes, one of them only on intratumoural or intraepithelial lymphocytes and four of them including both stromal and intratumoural components. Details of the methodology for TIL-Bs assessment in each study can be found in Appendix A. Multivariate analyses of TIL-Bs as an independent prognostic factor were conducted in all but two studies which focused only on the role of TIL-Bs in predicting response to neoadjuvant chemotherapy (NACT); most of them (n = 12) evaluated the prognostic effects of TIL-Bs on disease-free survival (DFS)/disease-free interval (DFI)/recurrence-free survival (RFS), whilst four of them studied breast cancer-specific survival (BCSS) and five of them evaluated overall survival (OS). Five studies examined the association between TIL-Bs and response to NACT.

#### 3.1.2. Tissue Sample

Studies were equally split between the use of full-face tissue section and tissue microarray (TMA) in evaluating TIL-Bs, with a tendency for those with a larger cohort to choose TMA over full-face sections. Five out of the seven studies with less than 200 patients used routine histopathological sections, whilst five out of the seven studies with more than 200 patients resorted to TMA. Originally designed as a high-throughput approach for researchers to assess protein or gene expression on hundreds of samples simultaneously [44], TMAs are much more cost- and time-efficient compared with conventional, full-face, single-sample tissue sections when assessing large numbers of tissue samples [45]. Despite this, there has been concern about tumour heterogeneity and sampling bias in the use of TMA over full-face sections [46]. In this review, all of the seven studies that used TMA used two or three tumour cores in the construction of TMA, with a selective sampling of tumour or tumour-rich areas. In their study with 338 patients, Mohammed et al. (2013) compared the scores of CD20+ and CD138+ TIL-Bs in full-faced sections with those in TMA for 40 patients and concluded that they were in excellent agreement with correlation coefficients of 0.99 and 0.80, respectively [39]. Details of tissue selection in TMA preparation for each of these studies have been tabulated in Table 2.

#### 3.1.3. Antibody and Counting Strategy

Among the studies that utilised antibodies to identify B cells (n = 13), all but one used the monoclonal anti-CD20 antibody from Dako (n = 9) or others (n = 3), whereas the remaining one used the polyclonal anti-CD20 antibody from Lab Vision. There were four studies that employed additional markers, anti-CD38 (n = 2) and/or anti-CD138 (n = 3), to assess the prognostic significance of PCs. One study relied on histo-morphological assessment alone in the identification of PCs.

Scoring of immuno-stained slides was then performed either manually (n = 9) or digitally using various software packages (n = 3). For those where the assessment was done manually, most of them specified at least two investigators being involved (n = 7). A variety of parameters were used to characterise TIL-Bs abundance, including the number of labelled cells within a specified area (n = 7), the percentage of labelled cells among total TILs (n = 4), as well as the percentage of intratumoural or stromal area occupied by labelled cells (n = 3). Most studies categorised the continuous variable into discrete groups for analysis (n = 11), where each tumour was classified as positive/negative or high/low for TIL-Bs, using the median or mean within the cohort (n = 5) or a threshold determined by statistical programming as the cutoff (n = 3). In three studies, the method of cutoff determination was not specified. Details of the TIL-Bs counting strategy in each study have been tabulated in Table 3.

### 3.2. TIL-Bs Abundance and Breast Cancer Subtype

Previous retrospective studies have shown that the abundance and clinical relevance of TIL-Ts depends in part on the intrinsic molecular subtypes of breast cancer [47,48,49,50,51]. More specifically, the density of all phenotypes of TIL-Ts is consistently lower in the luminal subtypes than in the non-luminal ones, which include the HER2-enriched and TNBC subtypes [47,48,49,50,51], implying that the latter groups are likely to be more immunogenic. In these latter subtypes, positive prognostic and predictive associations of TIL-Ts, or TILs as a general population, have been described in the contexts of NACT and adjuvant chemotherapy (ACT) for endpoints including complete pathological response (pCR) [52,53,54,55], recurrence-free survival (RFS) and overall survival (OS) [56,57,58,59,60]. Among the fourteen cohorts reviewed in this analysis, only three specifically compared the magnitude of TIL-Bs in TNBC versus non-TNBC subtypes [34,37,38]. Among them, two identified a significantly higher density of TIL-Bs in TNBC (*p* = 0.08 and *p* = 0.006, respectively) [37,38]. In studies which did not explicitly compare TNBC against non-TNBC, higher TIL-B counts were also found to be associated with higher tumour grades (n = 7), which reflect higher degrees of nuclear pleomorphism, higher mitotic counts and proliferation rates [61], and thus a greater likelihood of mutational events among tumour cells. In addition, Miligy and colleagues (2016) reported higher numbers of peritumoural and paratumoural B lymphocytes not only in invasive tumours with higher grades but also in tumours with necrosis (*p* = 0.05 and *p* = 0.03, respectively) [36]. It was speculated that the presence of necrosis could be associated with the release of damage-associated biomolecules such as adenosine triphosphate (ATP) into extracellular spaces, which could subsequently result in the recruitment of immune cells, including B lymphocytes, into the TME [62]. A similar association was seen in ductal carcinoma in-situ (DCIS) [36], which was consistent with the findings by Campbell et al. (2017), who observed higher CD20-positive TILs in high-grade DCIS with comedo necrosis than low-grade DCIS [62].

Interestingly, although Arias-Pulido and colleagues (2018) were not able to identify a statistically significant correlation between TIL-B density and TNBC, which might be related to the relatively low number of TNBC cases included in their study (41 TNBC vs. 175 non-TNBC), they reported a significant association between CD20+ TILs and programmed death protein-1 (PD-1) positive TILs and programmed death-ligand 1 (PD-L1) positive tumour cells [34], which could possibly reflect an orchestrated adaptive immune response within the TME, where activation of B lymphocytes leads to upregulated expression of checkpoint proteins among immune cells as an intrinsic mechanism of modulation and among tumour cells as a strategy to evade the immune system, a finding that supports the exploration of anti-PD-1/PD-L1 therapy in breast cancer patients.

### 3.3. TIL-Bs and Survival Outcome in Breast Cancer

#### 3.3.1. Clinico-Pathological Factors

Twelve studies evaluated the prognostic effect of TIL-Bs in breast cancer, including a total of 3521 patients [30,31,32,33,34,35,36,39,40,41,42]. The prognostic effect was positive in seven [30,31,32,33,34,35,41], negative in three [36,39,40] and neutral in two [42,43]. Among the seven studies which showed positive prognostic associations, increased TIL-Bs were related to better prognosis in terms of DFI/DFS/RFS (n = 6), OS (n = 4) and BCSS (n = 1). Such association was seen in studies that recruited exclusively TNBC and/or HER2-enriched cases (n = 4) [30,31,32,35], as well as in cohorts that included invasive breast cancer of non-specified, non-TNBC subtypes (n = 3) [33,34,41]. Among the latter three, a study by Mahmoud et al. (2012) showed that increased numbers of TIL-Bs were associated with better survival in TNBC, independent of tumour size and nodal stage (*p* < 0.001) [41]. Similarly, higher TIL-B counts were associated with better BCSS in HER2-enriched tumours, independent of tumour nodal stage and vascular invasion (*p* = 0.023) [41]. However, when tumour grades were taken into consideration, higher numbers of TIL-Bs have associated with better BCSS only in grade 3 tumours (*p* < 0.001) but not in grade 1 or grade 2 tumours [41]. Likewise, total B cell count was associated with better prognosis in estrogen receptor (ER)-negative tumours (*p* = 0.008) but not in ER-positive tumours (*p* = 0.261) [41]. On the other hand, Xu et al. (2018) and Arias-Pulido et al. (2018) reported that high CD20+ cell count was a favourable prognostic factor (*p* = 0.004 and *p* = 0.005 respectively) independent of tumour stage, histological grade, hormone receptors status and HER2 status [33,34]. Despite the varied findings among different tumour grades and ER statuses, the overall findings were supportive of a positive correlation between TIL-Bs and improved survival in invasive breast cancer, and the favourable effect was present irrespective of disease stage, which is dependent on tumour size and nodal status [63].

#### 3.3.2. TIL-Bs Subpopulations

Interestingly, Mohammed and colleagues reported in both of their studies in 2012 and 2013 the association between increased TIL-Bs and poorer BCSS. In both studies, increased tumour PC infiltrate, defined by morphological assessment under the microscope or CD138+ immunohistochemical positivity, was independently associated with poorer cancer-specific survival (*p* < 0.001) among patients with primary operable invasive ductal breast cancer [39,40]. In the latter study, which employed IHC for B-cell subtyping, increased CD8+ T-cell infiltrate/tumour CD138+ B-cell (i.e., PC) infiltrate ratio was also independently associated with improved cancer-specific survival (*p* < 0.001), whilst there was no significant association between CD20+ B-lymphocytic infiltrate and cancer survival [39]. The negative association was seen in both ER-positive and ER-negative subgroups and thus did not seem to be correlated with hormone receptor expression. The two other studies in Japan and the UK by Kuroda et al. (2021) and Miligy et al. (2016), who also subtyped TIL-Bs and specifically evaluated the prognostic value of CD138+ B-cells, did not find such a negative association. The latter study by Miligy and colleagues (2016) reported instead a negative correlation between TIL-Bs (defined by CD19 and/or CD20 positivities and CD138 negativity) and RFS. They found that for both pure DCIS cases and DCIS cases mixed with invasion, decreased density of TIL-Bs tended to be associated with better RFS (*p* = 0.008 & 0.04 and *p* = 0.01, respectively) [36]. The negative association between TIL-Bs and survival in DCIS is remarkable, but considering the limited number of patients in Miligy’s study (n = 80), expansion of similar work on a larger series of patients will help clarify the issue. Importantly, irrespective of breast cancer subtypes, the role of PCs is likely to be more complex than first considered, and further research is needed to shed light on their prognostic value in breast cancer.

#### 3.3.3. TIL-Bs Locations

Another important parameter in the study of TILs is the tumour region to be analysed. TILs are located within the intratumoural and stromal compartments [64]. Intratumoural TILs actively infiltrate tumour cell nests with no intervening stroma and have direct cell-to-cell contact with carcinoma cells, while stromal TILs are dispersed in the stromal tissue adjacent to the tumour cells that are still considered part of the malignant tumour, and do not come into direct contact with carcinoma cells [55]. The same holds true for all components of TILs, including TIL-Bs. All studies included in this review assessed the number of TIL-Bs by examining the H&E stained histological slides via light microscopy, with intratumoural or stromal areas specified in eight of them [30,32,33,34,36,38,41,43]. Four studies reported prognostic results for intratumoural TIL-Bs [30,32,36,41]; of these, three reported a positive prognostic effect [30,32,41], and one found no prognostic association [36]. Six studies reported prognostic results for stromal CD20+ TILs [30,32,33,34,36,41]; of these, five showed a positive prognostic effect [30,32,33,34,41], and one found a negative association [36]. In the latter, Miligy and colleagues (2016) revealed that for pure DCIS cases, low numbers of peritumoural and paratumoural B lymphocytes had a longer RFS (*p* = 0.008 and *p* = 0.04, respectively); similarly, for DCIS cases with an invasive component, less dense peritumoural B lymphocytes were associated with a longer RFS (*p* = 0.04) [36]. The studies by Mahmoud et al. (2012) and Miligy et al. (2016) were the only two studies in this review where stromal TIL-Bs were further compartmentalised with assessment areas clearly delineated. In the former, stromal TIL-Bs were categorised as distant stromal or adjacent stromal, depending on whether they were more than or within one tumour cell diameter away from the tumour [41]. In Miligy’s study (2016), the population was divided into peritumoural and paratumoural, with the former being defined as stromal TIL-Bs that were less than 0.5 mm from DCIS or invasive tumours and the latter as those that were between 0.5 mm to 1 mm away from the DCIS or invasive tumour36. Both subgroups of stromal TIL-Bs were implicated in these two studies. Six cohorts made no distinction between intratumoural and stromal TIL-Bs [31,35,37,39,40,42], with three of them non-specifically describing the area being assessed as “invasive front” [42] or “invasive margin” [39,40] and two of them generalising the site as “tumour area” [35,37]. Among them, two reported a positive prognostic effect [31,35], one found no association [42] and two found a negative association [39,40]. The remaining one reported a positive correlation with response to NACT [37].

In the study of TILs, the initial thinking was that intratumoural TILs, which are in direct contact with carcinoma cells, might be more important biologically and, therefore, more useful for the assessment of clinical relevance [65]. Subsequent studies have suggested that stromal TILs could be a parameter of equal significance, if not of even greater importance, as the two populations usually parallel each other, and stromal TILs almost always outnumber intratumoural TILs [66]. Findings in this review for TIL-Bs appear to be in line with this view, as no apparent differences were found in the direction of the prognostic effect regardless of whether intratumoural versus stromal TIL-Bs were assessed. Looking from another perspective, stromal TIL-Bs were implicated in all six studies where they were specifically evaluated and significant prognostic associations, positive or negative, were identified.

### 3.4. TIL-Bs and Response to Chemotherapy

The relationship between TILs and response to chemotherapy in solid tumours has attracted considerable attention in the past decade. Results from a number of large randomised trials have suggested that increased levels of TILs in breast cancer were associated with higher rates of complete pathological response (pCR) and greater benefits from chemotherapy, with no substantial differences between histological subgroups [53,57,59,66,67,68]. However, thus far, research efforts have been overwhelmingly focused on CD4+ and CD8+ T lymphocytes or TILs in general, with a remarkable paucity of experimental or clinical data on the significance of B lymphocytes specifically.

Five retrospective studies included in this review evaluated the predictive value of TIL-Bs in breast cancer [34,35,37,38,43], with the majority of them (n = 4) [34,35,37,38] showing a positive correlation between levels of TIL-Bs and rates of pCR. In all these five studies, the occurrence of pCR was identically defined as the absence of any residual invasive cancer in the breast and regional lymph nodes (ypT0/Tis, ypN0). Similar regimes of NACT were given, comprising a combination of taxanes and anthracyclines. Whilst Song and colleagues (2016) recruited only TNBC patients for their study, in which they found CD20+ cell density was significantly correlated with pCR (*p* = 0.037) [35], the rest included a mix of intrinsic breast cancer subtypes. In multivariate logistic regression analysis, Garcia-Martinez et al. (2014) confirmed an independent predictive value of a high baseline CD20+ TIL population for improved pCR (*p* = 0.005) [37] among a cohort of 108 patients comprising a mix of TNBC, HER2-enriched and luminal breast cancers. Similarly, Brown and colleagues (2014) found that among a mixed cohort of 95 patients with TNBC and non-TNBC, CD20+ TILs independently predicted pCR (*p* = 0.019), and more specifically, the rate of pCR was 5.5 times higher among high CD20 expressors [38].

Arias-Pulido et al. (2018) moved further to evaluate the associations of “CD20+ TILs plus PD-L1+ TILs” with pCR. Interestingly, while “positivity” in PD-1+ TILs alone was not associated with pCR, “co-positivities” of “CD20+ TILs plus PD-1+ TILs” and “CD20+ TILs plus PD-L1+ tumour cells” were significantly associated with high pCR rates (*p* = 0.04 and *p* = 0.005 respectively) [34]. This finding is intriguing and, at first sight, counter-intuitive, given that PD-1 and PD-L1 are inhibitory immune checkpoint molecules which serve to dampen an activated immune response. A plausible explanation is that the activation of B lymphocytes results in upregulated expression of inhibitory immune checkpoint proteins, including PD-1 on immune cells and PD-L1 on tumour cells as negative feedback [69], which is captured in one H&E section as a snapshot of the tumours at one specific point in time. In reality, the TME is immensely dynamic; the magnitude and net effects of the cellular and humoral immunity are dependent on the functional status and spatial relationships between different components, which cannot be truthfully revealed by immuno-histological assessments alone.

West et al. (2011) were the only group in this review which reported no statistically significant difference in pCR rates between CD20-high and CD20-low cases [43]. Although no data was provided on the intrinsic breast cancer subtypes recruited, since all patients within their cohort were ER-negative, it could be inferred that their breast cancer markers expression patterns could theoretically be: “ER-/PR-/HER2+”, “ER-/PR-/HER2-“, “ER-/PR+/HER2+” or “ER-/PR+/HER2-”. The first two patterns represent the HER2-enriched and the TNBC subtypes, respectively. It was reported that only three in the cohort were PR+; therefore, the majority (n = 110) were either HER2-enriched or TNBC, the biologically aggressive subgroups which were also included in the other four studies. Remarkably, West and colleagues only counted intraepithelial TIL-Bs in their assessment, defined as CD20+ lymphocytes within tumour cell nests or in direct contact with tumour cells [43], and disregarded the stromal population. Although it is not clear whether the lack of predictive value of TIL-Bs in their study could be attributed to this different scoring methodology, it highlights the challenge in comparing findings across limited studies where assessments of TIL-Bs were not standardised. The study would also have been much more informative if the stromal component had been included and separately evaluated in the analysis.

## 4. Discussion

### 4.1. Summary of Findings

Information on the clinical significance of TIL-Bs in breast cancer has thus far been limited in comparison with TIL-Ts. This is related to a multitude of factors which include the long-held dogma that breast cancer is not as immunogenic as other tumours like melanoma, colorectal cancer and NSCLC [70,71,72,73], the relative abundance of TIL-Ts in the TME of many solid tumours [51,74,75], as well as the deep-rooted notion that central memory and effector memory T lymphocytes confer anti-tumour immunity through cell-mediated cytotoxicity and cytokine-mediated mechanisms [76,77,78,79,80,81]. Primary studies included in this review showed that generally, when identified by CD20 immunohistochemical stain for TIL-Bs, the presence of B lymphocytes infiltrating mammary carcinoma was favourable for OS, DFS/RFS and BCSS [30,31,32,33,34,35,41], with a minority identifying no significant association [39,42,43] or negative associations with the malignancy [36]. The fact that these positive prognostic effects were independent of the presence or magnitude of T cell infiltration suggests that not only is the CMI important in breast cancer prognosis, but the humoral system may also contribute to a separate favourable effect. In addition, among studies that showed a positive prognostic value of TIL-Bs, no significant difference in clinical outcome was observed between patients with early-stage or late-stage disease [30,32,34,35,41], underscoring the biological relevance of humoral immunity across all stages of breast cancer. Whilst the independent prognostic value of TIL-Bs among TNBC and HER2-enriched breast cancers were more consistently reported [30,31,32,35], studies with mixed TNBC and non-TNBC subtypes showed more varied findings [33,34,41], although most studies indicated that the prognostic relevance was independent of hormone receptors status and HER2 status.

Unlike invasive breast cancer, where the presence of high TIL-Bs is linked with better prognosis, this did not appear to be the case for in-situ cancer (DCIS), and early invasive disease [36], insofar as this review has shown. Previous research has suggested that as breast cancer progresses from in-situ to invasive disease, there are genetic changes in the TME, including inflammatory cells and stromal fibroblasts, with the former displaying different interleukin signalling patterns and the latter transforming to a myofibroblastic phenotype with increased deposition of extracellular collagen matrix [82,83], suggesting that the functional status, spatial interaction and prognostic value of B cells may correspondingly change in the course of tumour development.

With regards to the location of TIL-Bs, the included studies indicated that both stromal and intraepithelial/intratumoural infiltrates could be associated with a favourable prognosis in invasive breast cancer [30,32,33,34,36,38,41], but the findings appeared to be less consistent with the intraepithelial/intratumoural component in an earlier study [43]. Indeed, although not relating to B cells specifically, there is recent evidence which suggests that intraepithelial TILs in direct contact with the tumour cells may actually have a different molecular phenotype than stromal TILs more distant from the tumour proper [84].

In addition to the favourable prognostic value of CD20+ B cell infiltration, the role of tumour-infiltrating plasma cells, defined immunohistochemically as CD38+ or CD138+ cells in most studies, has been explored [30,32,36,39,40]. Unlike the relatively consistent evidence of the beneficial value of CD20+ B cell infiltration, the analyses of plasma cells have yielded more conflicting results, with data showing a favourable prognosis [30,32], no prognostic value [36] and adverse prognosis [39,40]. Clinicopathological parameters, including intrinsic breast cancer subtypes, could not account for the differences observed in these studies. What appears to be significant, instead, is that the positive prognostic values were only reported when plasma cells were identified as CD38+ cells [30,32]; where a neutral or negative association was identified, plasma cells were detected by anti-human CD138 antibody (clone MI15) [36,39]. These findings highlight the fact that appropriate immunohistochemical markers targeting different B cell phenotypes with varying specificities could be vital for evaluating the prognostic significance among TIL-B subpopulations.

Increasing TIL-Bs have also been shown to be predictive of pCR to NACT, not only in TNBC but also in non-TNBC subtypes, including ER-positive breast cancer in the context of neoadjuvant anthracycline and taxane-based chemotherapy [34,35,37,38]. Similar to the favourable prognostic value, the positive predictive role of CD20+ TILs was independent of CD3+ and/or CD8+ TILs [34,35,38], lending support to the significance of B cell immunity on top of CMI in breast cancer. The combined study of immune checkpoint proteins (PD-1 and PD-L1) and B lymphocytes further revealed a plausible positive predictive synergy in using CD20+ TILs and PD-L1+ tumour cells in selecting patients for neoadjuvant treatment [34].

### 4.2. Pertinent Issues and Gaps in Knowledge

This review shows accumulating, albeit hitherto limited, evidence which supports the independent prognostic and predictive value of TIL-Bs in breast cancer. Importantly, there are a number of pertinent issues and areas of uncertainty that could be addressed in the future to further decipher and clarify the role of infiltrating B lymphocytes in breast cancer, and indeed, all other solid cancers, in the quest for novel immunotherapies. These outstanding questions pertaining to the practical issues encountered by investigators in scoring TIL-Bs, as well as areas of an incomplete understanding of the fundamentals of B cell biology in cancers.

#### 4.2.1. Scoring Methodology

Studies included in this review showed variations in their strategies for assessing TIL-Bs, from sample preparation and area selection to the actual enumeration and determination of cutoff for analysis. Although some studies using TMAs and well-annotated clinical datasets suggested that results could be largely concordant with those using full-face tissue sections [85,86,87] and that tissue heterogeneity could be overcome by punching multiple cores from the same tumour, intratumoural heterogeneity remains an issue that should not be overlooked, particularly in light of the distribution pattern of infiltrating B lymphocytes in some solid tumours, including breast cancer. A previous study using full-face tissue sections described a clustered pattern of peritumoral B cell infiltrates in up to 52% of breast cancer [88]. Although all studies in this review resorting to TMA described using a range of two to three tumour cores in the construction of TMA, theoretically, this would have only increased the proportion of the total tumour examined minimally. While TMAs may be a good option for future studies, particularly for the rapid evaluation of large clinical cohorts, more studies would be needed to examine the agreement between TMA cores and full-face sections from different paraffin blocks of the same tumour and to reach a consensus on the optimal number of cores that should be taken.

Until recently, there has been no official guideline on the assessment of TILs in solid tumours. Thanks to the International Immuno-Oncology Biomarker Working Group, a recommendation was first formulated in 2013, which sought to standardise methodology and scoring systems for the integration of information about TILs into future research [89,90]. While the guidelines define the stromal and intra-tumoural area for assessment, no recommendation is provided as to how the continuous variable should be categorised and how a clinically relevant cutoff could be determined [89]. Most studies in this review either dichotomised or trichotomised TIL-B levels for survival or outcome analyses, but the criteria of cut-points were variable. This problem could be particularly profound in the study of TIL-Bs, which in comparison to TIL-Ts, usually show a much narrower range of abundance [20], so what was classified as B cell-rich in one study could have been categorised as B cell-poor in another, and vice versa. Therefore, to generate reliable data on TIL-Bs in large-scale clinical trials in the future, standardised methodology and scoring systems would be imperative.

#### 4.2.2. Characterisation of TIL-Bs Subpopulations

The negative prognostic role of PCs revealed by some studies is remarkable. Nevertheless, the use of CD38 and CD138 as markers for PCs could be controversial. Although well-known to be expressed by B cells with plasmacytic differentiation [91,92], both molecules can be present in other cell types as well. CD38 is a glycoprotein found almost ubiquitously on the surface of multiple immune cells, including PCs, activated B cells, GC B cells, T cells, NK cells, macrophages and DCs [93,94,95,96,97], whereas CD138, also known as syndecan-1, is an integral membrane proteoglycan [98] expressed not only in PCs but also in epithelial and stromal cells [99], including even breast carcinoma cells [32,100]. Such antibody cross-reactivities could potentially confound the evaluation of plasma cell levels in the tumour bed. Already in use in some cancer research, the problem might be circumvented by multiplexed IHC, which makes possible the simultaneous detection of multiple markers on a single tissue section [101], thereby allowing different cell populations to be more precisely classified by providing multi-layered information on the phenotypic characteristics of each individual cell. Coupled with multispectral imaging, a more comprehensive view of the spatial distribution and architectural composition of immune cell subsets could also be provided [101,102], potentially offering further insights into the cellular landscape within the TME.

Similarly, almost all studies in this review used anti-human CD20 antibody (clone L26) to highlight B cells, but the use of this nearly universal B cell antigen would not allow the differentiation between anti-tumour B cells and regulatory B cells (Bregs), which can play an immunosuppressive role and drive tumour growth [103,104]. Unlike Tregs, which are commonly identified by their expression of surface markers like CD25 and Foxp3 [105], the phenotypic profile of Bregs is poorly defined, with some studies describing CD19+CD20+CD27−CD21− and CD19+CD20+CD5+ B cells as the pathogenic populations [25]. In the research setting, they are characterised by the secretion of immunosuppressive cytokines, including IL-10 and TGF-β [26]. It is, therefore, possible that tumours being categorised as high or positive in TIL-Bs by CD20 immunohistochemical staining were actually rich in Bregs, leading to the conclusion that CD20+ cells have no relevance or even negative prognostic value in some studies. Future research efforts should therefore seek to identify more specific markers that could allow the differentiation of Bregs from anti-tumour B cells. A more sophisticated analytical approach that combines multiple platforms, such as multiplexed IHC and in-situ proteomic analysis, whereby protein products related to specific TIL-B subtypes could be identified in situ without the need for specific antibodies [106], could be of immense value.

### 4.3. Future Directions

In this era of precision cancer medicine, immunotherapy never ceases to be the focus of cancer therapeutics. This is no exception in breast cancer. In 2019, atezolizumab, an anti-PD-L1 monoclonal antibody, was approved in combination with nab-paclitaxel chemotherapy as first-line treatment of patients with locally advanced or metastatic TNBC whose tumours were positive for PD-L1 expression (≥1%) in immune infiltrate, and the Ventana PD-L1 (SP142) antibody had been approved as the complementary diagnostics by the Food and Drug Administration in the United States [107]. The approval was subsequently withdrawn in 2021 [108], suggesting the insufficiency of using PD-L1 alone in selecting breast cancer patients for immunotherapy. The positive predictive synergy between CD20+ TILs and PD-1+ immune cells or PD-L1+ tumour cells revealed by Arias-Pulido et al. (2018) could serve as a springboard to stimulate more large-scale randomised controlled studies to assess the potential use of TIL-Bs as a component of a multi-biomarkers panel for the identification of patients who may benefit from immunotherapy.

Finally, to fully unravel the clinical significance of TIL-Bs in our pursuit of an immune-based armamentarium against breast cancer, or cancer in general, it is insufficient to garner information only on the density of TIL-Bs within the tumour bed. Details regarding their spatial locations, functional status and dynamic interactions with other components within the complex ecosystems [109] of the TME will all go a long way to contributing to a meticulous and systematic characterisation of their mechanisms of action. Concepts related to tertiary lymphoid structures (TLS) and Crohn’s-like reactions, which are better described in colorectal cancer and interpreted as an immune-mediated anti-tumour effect [110], can also be characterised and studied in breast cancer. To this end, novel genomic techniques, including scRNA-seq and spatial transcriptomics, are poised to be transformative in the near future.

### 4.4. Review Limitations

The generalisability of the findings in this review article could be limited due to shortcomings related to the literature review process.

The major limitation is that the database search, relevance assessment and studies inclusion and exclusion process were conducted single-handedly by the author of this review, owing to practicality amid the COVID-19 pandemic. The major risk of a one-person review is subjectivity, which could have been minimised through independent screening by multiple reviewers. Multiple screening reduces the chance of systematic bias and random errors, and where there is any lack of consensus among reviewers with regard to eligibility, consensus could be reached through discussion or consultation with experts.

A second limitation is the restricted search strategy. To make the yields more manageable and the review more focused, the search was restricted to articles that had been written in English. The studies identified were limited, and the systematic literature review process, as well as the write-up of the manuscript itself, is, to a certain extent, more akin to that of a critical review. Nevertheless, it is still hoped that the work being presented herein could serve to bring up pertinent issues that could be addressed in future research efforts pertaining to TIL-Bs in breast cancer.

## 5. Conclusions

This article has presented a systematic literature review of what is known thus far about the clinical significance of TIL-Bs in breast cancer.

From this review, it is apparent that there is accumulating evidence of an independent prognostic and predictive value of TIL-Bs in breast cancer. A higher density of TIL-Bs confers a more favourable prognosis and predicts better response to chemotherapy across different stages of breast cancer, particularly in high-grade tumours with more aggressive behaviour. In light of the major findings of this thesis, future suggestions with regard to the potential areas of development in the study of TIL-Bs were discussed. While standardised methodology and more specific immunostaining panels are indispensable in the detection and quantification of different B cell subpopulations, novel genomic technologies promise to allow further insights to be gained into their functional role and cellular dynamics. More large-scale research and multi-centre studies with larger cohorts will serve to reinforce the potential of TIL-Bs or the subpopulations therein as a therapeutic target, prognostic marker and predictive factor in different subtypes of breast cancer.

Overall, having a contemporary overview is key to identifying limitations in the current body of knowledge and developing new directions for future research. Investment in these potential objectives will pay dividends in the form of improved cancer therapeutics and, ultimately, patient outcomes.

## Figures and Tables

**Figure 1 cancers-15-01164-f001:**
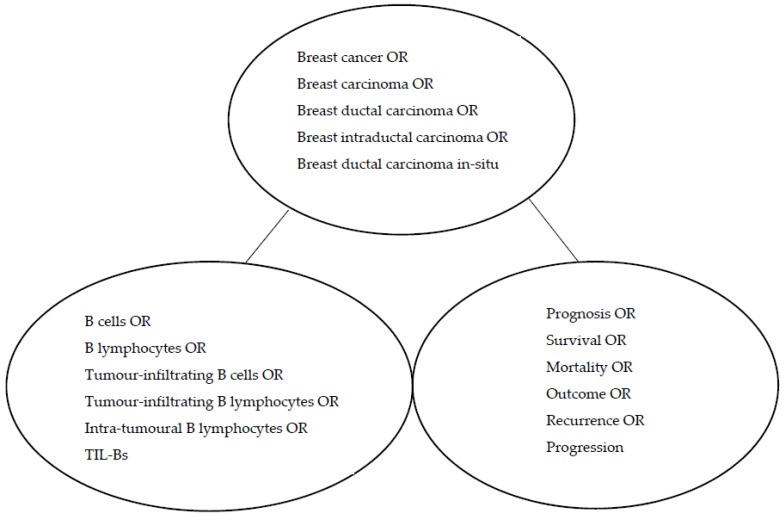
Boolean search with keywords and their synonyms.

**Figure 2 cancers-15-01164-f002:**
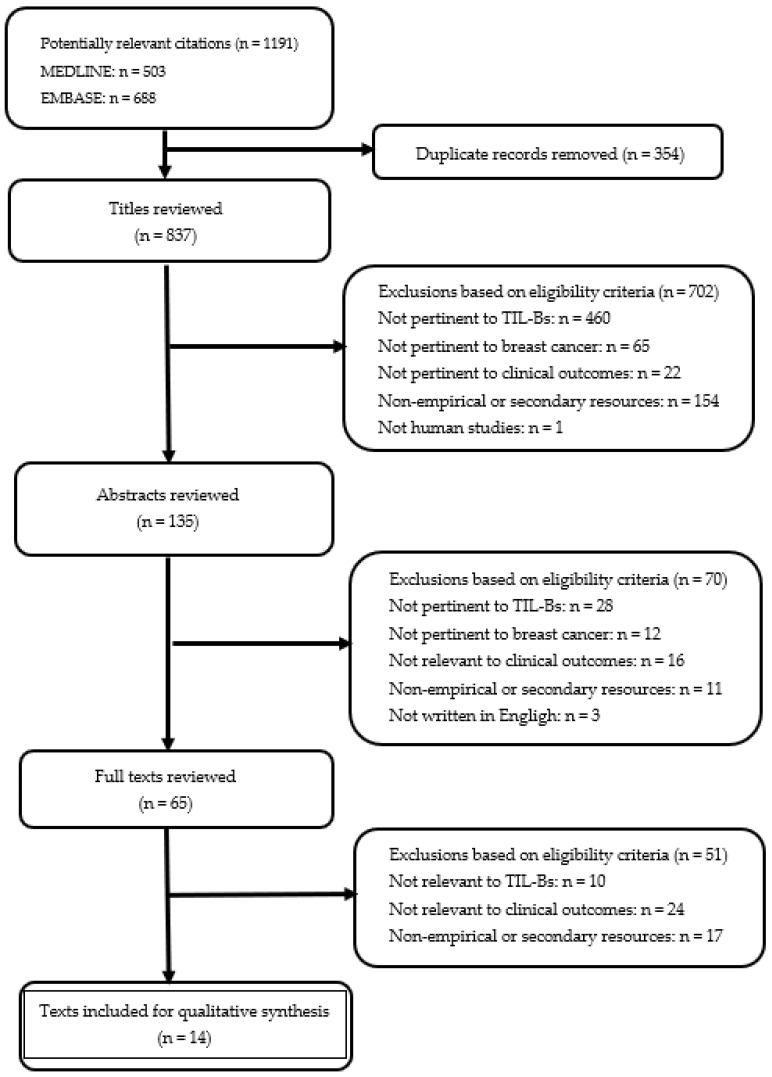
Flowchart describing the literature inclusion process.

**Table 1 cancers-15-01164-t001:** B cell markers.

B-Cell Subtype	Immunohistochemical Markers
Naïve B cell	CD19+, CD20+, IgM+IgD−, CD38−/−
Naïve activated B cell	CD19+, CD20+, IgM+, IgD+, CD38+
Germinal centre B cell	CD19+, CD20+, IgM+/−, IgD+, CD38++
Plasmablast	CD19+, CD20−, CD38++, CD27++, IgD-, IgM/G/A/E+
Plasma cell	CD19+/−, CD20−, CD38++, CD138+, CD27+ IgD−, IgM/G/A/E+
Memory B cell	CD19+ CD20+ CD38−, CD27+, IgD−, IgM/G/A/E+

**Table 2 cancers-15-01164-t002:** Tissue selection in studies that used TMA for assessing TIL-Bs.

Year	Authors	Selection of Tissue for TMA Construction
2018	Yeong et al. [32]	Two to three representative tumour cores (with >50% tumour area) of 1 mm diameter
2018	Arias-Pulido et al. [34]	Two 1.5 mm cores
2014	Garcia-Martinez et al. [37]	Two 2 mm cores with predominantly tumour areas selected by a pathologist
2013	Mohammed et al. [39]	Three 0.6 mm cores taken from the tumour-rich area
2012	Mahmoud et al. [41]	Three 0.6 mm cores obtained from the periphery of the tumours
2012	Eiro et al. [42]	Two 1.5 mm cores of invasive front and two 1.5 mm cores of tumour centre
2011	West et al. [43]	Two 0.6 mm cores from central cellular areas of tumour

**Table 3 cancers-15-01164-t003:** TIL-Bs assessment strategy in each study.

Year	Authors	Manual vs. Digital Scoring	Parameter	Determination of Cutoff for Categorization
2021	Kuroda et al. [30]	Manual	Number of stained cells in intratumoural or stromal area at 400× magnification	Median value of cohort
2019	Garaud et al. [31]	Manual, 2 trained pathologists	Percentage of stained cells in total TILs	%FINDCUT SAS macro
2018	Yeong et al. [32]	Manual, 2 pathologists	Percentage of intratumoural or stromal area occupied by stained cells at 400× magnification	Median value of cohort
2018	Xu et al. [33]	Manual, 2 investigators	Mean number of stained cells in 3 selected stromal areas (hotspot)	Mean value of cohort
2018	Arias-Pulido et al. [34]	Manual, board- certified pathologists	Percentage of stained cells in total TILs	Not specified
2016	Song et al. [35]	Digital, NuclearQuant module (3DHISTECH Ltd., Budapest, Hungary)	Number of stained cells in tumour area	Continuous variable
2016	Miligy et al. [36]	Digital, module not specified (3DHISTECH Ltd.)	Percentage of intratumoural or stromal area (hotspot) occupied by stained cells at 400× magnification	X-tile software
2014	Garcia- Martinez et al. [37]	Digital, Image J software (NIH, USA)	Number of stained cells per mm^2^	Continuous variable
2014	Brown et al. [38]	Digital, AQUA software	Percentage of tumour stroma that contained stained cells	Joinpoint software
2013	Mohammed et al. [39]	Manual, 2 observers	Mean percentage of stained cells in 4–12 fields at 400× magnification	Not specified
2012	Mohammed et al. [40]	Manual, 2 observers (including 1 pathologist)	Mean percentage of plasma cells in 4–12 fields at 400× magnification	Not specified
2012	Mahmoud et al. [41]	Manual, 3 investigators	Number of stained cells	X-tile software
2012	Eiro et al. [42]	Digital, analysis (Soft Imaging System, Germany)	Median number of CD20+ cells in 5 fields of an area of 1 mm^2^	Median value of cohort
2011	West et al. [43]	Manual	Number of intraepithelial CD20+ cells within an area of 0.56 mm^2^	Median value of cohort

## Data Availability

No new data were created or analyzed in this study. Data sharing is not applicable to this article.

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
