# Peer review of "Clinical Significance of Tumour-Infiltrating B Lymphocytes (TIL-Bs) in Breast Cancer: A Systematic Literature Review"

_cancers, 2023, doi:10.3390/cancers15041164_

Round 1
Reviewer 1 Report
The submitted manuscript seems to be interested, however, I have some comments and questions:
The first question is about the originality of this work, because several recent reviews have been published, some of them were published in Nature Reviews Cancer.
The used figures to explain the mechanisms have a low resolution, the authors should improve the resolution of all used figure,
Author Response
1) There may be other systematic reviews in the literature studying tumor-infiltrating B lymphocytes, but only a handful of them focus on breast cancer, and even if they do, there's been no critical appraisal in these reviews on the methodology of the primary studies involved. This is what makes this systematic review unique and original.
2) The authors thank the reviewer for their comment and would like to highlight that they do not have access to profession illustration services. The figure is therefore as optimized and high a resolution as we can achieve. If this is not sufficient, we would need to remove the figure from the manuscript or ask that the journal provide support in improving this.
Reviewer 2 Report
The authors summarized the significance of tumor-infiltrating B lymphocytes (TIL-B) in breast cancers by reviewing recent publications.
The study is well designed and suitable for publication.
Please address following points:
1. The prognostic effect of TIL-B seems vary among the studies reviewed in the manuscript. Please further address the reason of inconsistence (patients' background including subtypes, evaluation methods and so on).
2. Germinal center is often seen in breast cancer tissues and therefore it is interesting to consider the significance of germinal center in breast cancer tissues. Is it mentioned in the previous immunohistochemical studies?
Author Response
1) The prognostic effects of tumor-infiltrating B lymphocytes show inconsistency in a minority of the studies reviewed. By and large, TIL-Bs appear to show positive positive prognostic effects on breast cancer. As suggested in the discussion, the inconsistency could stem from i) the study methodology like the types of antibody clones being used and ii) presence of different subtypes of B cells within the tumor bed which may have different functionalities and are yet to be individually quantified from the general B cell population.
Thus for this point, no changes have been made to the manuscript.
2) Germinal center reaction has been more well described in colorectal cancer and not in breast cancer. So it has exactly been suggested in the paper that more studies on this aspect will be of value in the future.
Thus for this point, no changes have been made to the manuscript.
Reviewer 3 Report
The authors in this review summarized studies that demonstrated the importance of tumor-infiltrating B lymphocytes in breast cancer. They have suggested future research directions for understanding the role of B lymphocytes in breast cancer. The review backs up the findings in the manuscript. It can, however, be improved for a better version of this manuscript.
Here are some suggestions for enhancements.
Please cite references 27 and 28 under the bracket on line 120.
Figure 1: Please elaborate on Figure 1. It's difficult to tell what role B cells play. It is helpful to the reader if it is explained in the figure.
Figure 2: Better visibility is required. It's unclear in the figure.
Figure 3: Please enhance the visibility.
Overall, the review is favorable. The review is well-written and includes a systematic literature review of what is currently known 637 about the clinical significance of TIL-Bs in breast cancer.
Author Response
Many thanks indeed for the positive feedback and suggestions. Please find below responses to the comments:
References 27 and 28 will be cited under brackets.
Figure 1 will be deleted from the manuscript given the unfavorable feedback from various reviewers
Figures 2 and 3 will be revised as suggested.